# Decision Feedback Modulation Recognition with Channel Estimation for Amplify and Forward Two-Path Consecutive Relaying Systems

**DOI:** 10.3390/s22166022

**Published:** 2022-08-12

**Authors:** Mohamed Marey, Maged Abdullah Esmail, Hala Mostafa

**Affiliations:** 1Smart Systems Engineering Laboratory, College of Engineering, Prince Sultan University, Riyadh 11586, Saudi Arabia; 2Department of Information Technology, College of Computer and Information Sciences, Princess Nourah bint Abdulrahman University, P.O. Box 84428, Riyadh 11671, Saudi Arabia

**Keywords:** modulation recognition, consecutive relaying, soft information

## Abstract

Automatic modulation recognition (AMR) is an essential component in the design of smart radios that can intelligently communicate with their surroundings in order to make the most efficient use of available resources. Throughout the last few decades, this issue has been subjected to in-depth examination in the published research literature. To the best of the authors’ knowledge, there have only been a few studies that have been specifically devoted to the task of performing AMR across cooperative wireless transmissions. In this contribution, we examine the AMR problem in the context of amplify-and-forward (AAF) two-path consecutive relaying systems (TCRS) for the first time in the literature. We leverage the property of data redundancy associated with AAF-TCRS signals to design a decision feedback iterative modulation recognizer via an expectation-maximization procedure. The proposed recognizer incorporates the soft information produced by the data detection process as a priori knowledge to generate the a posteriori expectations of the information symbols, which are employed as training symbols. The proposed algorithm additionally involves the development of an estimate of the channel coefficients as a secondary activity. The simulation outcomes have validated the feasibility of the proposed design by demonstrating its capacity to achieve an excellent recognition performance under a wide range of running conditions. According to the findings, the suggested technique converges within six rounds, achieving perfect recognition performance at a signal-to-noise ratio of 14 dB. Furthermore, the minimal pilot-to-frame-size ratio necessary to successfully execute the iterative procedure is 0.07. In addition, the proposed method is essentially immune to time offset and performs well throughout a broad range of frequency offset. Lastly, the proposed strategy beats the existing techniques in recognition accuracy while requiring a low level of processing complexity.

## 1. Introduction

The increasing growth in the number of end-users in wireless communication networks has raised the reliability and stability criteria that must be satisfied by both current and future networks. Providing such conditions is challenging and requires building intelligent and sophisticated networks to ensure high quality-of-service for the end-users and to help with agile management of wireless network resources [1]. Automatic modulation recognition (AMR) plays a significant role in the development of such networks for numerous military and civilian applications such as threat assessment, link adaption, software-defined radio, spectrum monitoring, and cognitive radio [2,3,4,5,6,7,8,9]. Table 1 provides definitions for acronyms used in this work.

The modulation information may be incorporated in each broadcast frame when adaptive modulation is used to mitigate the effects of channel distortion or to offer on-demand data rate service [10,11]. However, such a strategy is not spectrum efficient for real wireless networks due to transmitting extra control information. After the signal has been detected and before it has been demodulated, AMR algorithms demonstrate their significance by determining the kind of modulation present at the receiver in order to perform accurate signal demodulation without requiring any prior information. This results in a reduction in the signal overhead and an enhancement in the spectrum’s efficiency [12]. Another example that highlights the relevance of AMR is offered by cognitive radios, which permits unlicensed users to transmit their data over the bands that are currently being used by other legitimate users. An AMR algorithm is an essential part of the design of cognitive radios, and its primary function is to determine the kind of modulations that are already present in the signals that have been received [13,14,15]. This enables cognitive radios to communicate without causing harmful interference to users who are currently utilizing their bands.

AMR has recently found practical applications in wireless sensor networks, which have evolved as potential solutions to a wide variety of real-life problems [16,17]. A viable AMR technique for sensor networks should provide a fusion centre, which merges local information obtained and/or created by separate sensing nodes [18]. Fusion techniques of this kind may be used at the data, feature, and decision levels [19]. The fusion center receives the raw signal data directly from each sensor node in a data-level fusion process. Despite the fact that each sensor is only needed to do little processing, the data-level fusion method requires a substantial transmission overhead from each sensor to the fusion center. In a feature-based fusion methods, each sensor extracts features from the collected data and then communicates the recovered features to the fusion center, which needs all sensors to be closely synchronized with each other in order to make a final judgment. In decision-layer fusion processes, each sensor creates a local judgement based on the extracted characteristics, which is then sent to the fusion centre, which derives the global judgement based on the local choices. In general, combined signals from a group of geographically distributed sensors will produce a higher performance for modulation recognition than any one node would achieve on its own [20]. For example, deep residual network and long short time memory are adopted for modulation recognition with distributed sensors [21], yielding better recognition accuracy than a single convolution neural network.

Alternatively, cooperative diversity is a relatively recent category of wireless communication systems in which network nodes assist each other in relaying information to improve connection reliability, spectrum efficiency, system capacity, and transmission coverage [22,23,24]. As a result, cooperative communication seems to be similar to a distributed multiple-input multiple-output (MIMO) system that utilizes a virtual antenna array. Amplify-and-forward (AAF), decode-and-forward, and compress-and-forward are examples of the cooperative protocols that have been developed so far [25,26,27]. Full-duplex and half-duplex relaying are the two primary varieties of this communication technique. Full-duplex communications occur when a relay is able to simultaneously send and receive signals within the same frequency range. Even though the relay is able to eliminate interference between the transmitted and received signals, a slight error in the cancellation process can result in a significant decrease in performance due to the fact that the power of the received signal is significantly less than that of the transmitted signal [28,29]. Because of this, it is recommended to employ half-duplex relaying, in which the relaying node either broadcasts or receives information at the same time. Half-duplex, on the other hand, suffers from the drawback of reduced spectral efficiency, which results in a throughput drop of fifty percent [30]. One possible solution to deal with this challenge is to use two-path consecutive relaying systems (TCRS) [31,32,33,34]. To be more detailed, the source broadcasts its information to one of the relays and the destination at any given time slot, while the other relay transfers the information it got from the source during the time slot preceding it to the destination. Because of this, the source is able to send data in a continuous manner, which allows the spectral efficiency loss to be restored.

There has been a significant amount of effort put into the information theoretic study of a variety of cooperative systems in order to predict the rates that are practical, the capacity restrictions, the diversity-multiplexing tradeoff, as well as data extraction and channel identification (e.g., [31,35,36,37,38,39]). In spite of the significance of these discoveries, further research has to be conducted before cooperative systems may be put into reality. One of the most important difficulties that has to be tackled is determining how AMR is accomplished in cooperative systems. The development of AMR algorithms for cooperative systems has only been the focus of a relatively small number of research up to this time. Recently, the maximum-likelihood (ML) solution and cross-correlation functions were used in order to carry out AMR over decode-and-forward one-relay cooperative systems [40,41].

The following are the primary contributions of this study.

For the first time in the lecture, an AMR algorithm for TCRS using an AAF protocol is proposed in this work.Through the use of the expectation-maximization (EM) process [42,43], the maximum-likelihood (ML) principle is used to arrive at an estimate of the allotted modulation scheme.The proposed algorithm makes use of the a posteriori probabilities that are acquired by the data detector as a priori knowledge to create the a posteriori expectations of the broadcast symbols, which are utilized as training symbols.Instead of using different algorithms to predict the channel impulse responses that occur between the source and relays, the source and the destination, and the relays and the destination, we estimate a single parameter called the overall channel impulse response, which includes all these connections.

The remaining parts of the work are structured in the following manner. In Section 2, the cooperative system model under consideration is presented. The proposed AMR algorithm and its practical discussion are offered in Section 3 and Section 4. The simulation results with appropriate analysis are provided in Section 5. Finally, we conclude the work in Section 6.

## 2. System Model

We examine a wireless cooperative network that includes a source (*S*), a destination (*D*), and two half-duplex relays ($R_{1}$ and $R_{2}$). All nodes have a single antenna, and relays employ an AAF protocol. The source data are sent in the form of frames, and each frame comprises *W* packets. Each packet is made up of *C* data symbols, all of which are taken at random and autonomously from a modulation constellation known as Φ. The key idea of adaptive modulation is that the source makes adjustments to its modulation scheme on a frame-by-frame basis in accordance with the channel impulse response. These adjustments are made in order to maximize cumulative capacity and reducing power consumption while preserving a level of service that is satisfactory. A limited number of pilots are integrated into data symbols to initiate the recognition operation as we will show later on. The *w*th packet is expressed as $p^{\left(w\right)}=\left[p^{\left(w\right)}\left(0\right),\;p^{\left(w\right)}\left(1\right),\;\cdots ,p^{\left(w\right)}\left(N-1\right)\right]$, where $p^{\left(w\right)}\left(m\right)$ is the *m*th data symbol of *w*th packet. The channel coefficient from terminal ι to terminal κ is represented as $g_{\iota \kappa }$, where $\iota \in \left\{S,R_{1},R_{2}\right\}$ and $\kappa \in \left\{R_{1},R_{2},D\right\}$. Channel coefficients stay the same during a given frame, but they change independently from one frame to the next.

The transmission protocol is broken up into W+1 consecutive time slots that are all the same length, and during each slot, *S* sends independent *W* packets constantly. In the following description, we use the common assumption that the two relays are placed in close proximity to one another or at a more distant position [44,45]. In such a scenario, it is possible to employ a consecutive interference cancellation scheme at the relays. Because of this, the performance at the relays is quite similar to what it would be like if the inter-relay interference was completely eliminated. As observed in Figure 1, the procedure consists of two distinct stages: odd and even time slots.

At even time slots 2b: the source broadcasts the packet $p^{\left(2b\right)}$ to $R_{1}$ and *D* while $R_{2}$ sends the signal $s_{R_{2}}^{\left(2b\right)}$ to *D*. Here $s_{R_{2}}^{\left(2b\right)}$ is represented in terms of the preceding packet as [28]
(1)$$s_{R_{2}}^{\left(2b\right)}=A_{2}\left(g_{SR_{2}}p^{\left(2b-1\right)}+z_{R_{2}}^{\left(2b-1\right)}\right),$$
where $z_{R_{2}}^{\left(2b-1\right)}$ is the (2b−1)th noise vector at $R_{2}$ of length *N* coupled with the preceding source transmission and $A_{2}$ is the scaling factor provided as [29]
(2)$$A_{2}=\frac{1}{\sqrt{{\left|g_{SR_{2}}\right|}^{2}+\sigma_{n}^{2}}}.$$Therefore, the received signals at $R_{1}$ and *D* can be written as [31]
(3)$$r_{R_{1}}^{\left(2b\right)}=g_{SR_{1}}p^{\left(2b\right)}+z_{R_{1}}^{\left(2b\right)},$$
(4)$$r_{D}^{\left(2b\right)}=g_{SD}p^{\left(2b\right)}+g_{R_{2}D}s_{R_{2}}^{\left(2b\right)}+z_{D}^{\left(2b\right)},$$
where $z_{R_{1}}^{\left(2b\right)}$ and $z_{D}^{\left(2b\right)}$ are the noise components at $R_{1}$ and *D*, respectively. Using (Equation 1) into (Equation 4), one writes [39]
(5)$$r_{D}^{\left(2b\right)}=g_{SD}p^{\left(2b\right)}+{\bar{g}}_{R_{2}D}p^{\left(2b-1\right)}+{\bar{z}}_{D}^{\left(2b\right)},$$
where
(6)$${\bar{g}}_{R_{2}D}=A_{2}g_{R_{2}D}g_{SR_{2}},$$
and
(7)$${\bar{z}}_{D}^{\left(2b\right)}=z_{D}^{\left(2b\right)}+A_{2}g_{R_{2}D}z_{R_{2}}^{\left(2b-1\right)}.$$At odd time slots 2b+1, *S* conveys the packet $p^{\left(2b+1\right)}$ to $R_{2}$ and *D* while $R_{1}$ passes the information $s_{R_{1}}^{\left(2b+1\right)}$ to *D*. Here $s_{R_{1}}^{\left(2b+1\right)}$ is expressed in context of the previous packet as [35]
(8)$$s_{R_{1}}^{\left(2b+1\right)}=A_{1}\left(g_{SR_{1}}p^{\left(2b\right)}+z_{R_{1}}^{\left(2b\right)}\right),$$
where $z_{R_{1}}^{\left(2b\right)}$ is the (2b)th noise vector at $R_{1}$ and $A_{1}$ is the scaling factor given as [31]
(9)$$A_{1}=\frac{1}{\sqrt{{\left|g_{SR_{1}}\right|}^{2}+\sigma_{n}^{2}}}.$$The signals that are received at $R_{2}$ and *D* may be expressed as [38,39]
(10)$$r_{R_{2}}^{\left(2b+1\right)}=g_{SR_{2}}p^{\left(2b+1\right)}+z_{R_{2}}^{\left(2b+1\right)},$$
(11)$$r_{D}^{\left(2b+1\right)}=g_{SD}p^{\left(2b+1\right)}+g_{R_{1}D}s_{R_{1}}^{\left(2b+1\right)}+z_{D}^{\left(2b+1\right)},$$
where $z_{R_{2}}^{\left(2b+1\right)}$ and $z_{D}^{\left(2b+1\right)}$ represent the noise vector at $R_{2}$ and *D*, respectively. Using (Equation 1) into (Equation 11), one can show that [31]
(12)$$r_{D}^{\left(2b+1\right)}=g_{SD}p^{\left(2b+1\right)}+{\bar{g}}_{R_{1}D}p^{\left(2b\right)}+{\bar{z}}_{D}^{\left(2b+1\right)},$$
where
(13)$${\bar{g}}_{R_{1}D}=A_{1}g_{R_{1}D}g_{SR_{1}},$$
and
(14)$${\bar{z}}_{D}^{\left(2b+1\right)}=z_{D}^{\left(2b+1\right)}+A_{1}g_{R_{1}D}z_{R_{1}}^{\left(2b\right)}.$$

The main objective of this study is to create a modulation recognition algorithm that takes use of the distinct structure of the received signals at the destination, $r_{D}^{\left(2b\right)}$ and $r_{D}^{\left(2b+1\right)}$.

## 3. Proposed Algorithm

The subsequent description will highlight how the procedures of modulation recognition and channel estimation are carried out. We begin with the ML criterion and build a data-aided (DA) recognizer and estimator. After that, a soft information assisted recognizer and estimator is constructed with the help of the EM algorithm. The unique structure of the signals that have been received is used to provide more accurate results.

We establish the following notations for use in mathematical representation. Let us write $R=\left[r_{D}^{\left(0\right)},r_{D}^{\left(1\right)},\cdots ,r_{D}^{\left(W\right)}\right]$, $G={\left[g_{SD},{\bar{g}}_{R_{1}D},{\bar{g}}_{R_{2}D}\right]}^{†},$ and [46,47]
(15)$$P=\left[\begin{array}{ccc} p^{\left(0\right)} & 0_{1\times N} & 0_{1\times N}\\ p^{\left(1\right)} & p^{\prime }{}^{\left(0\right)} & p^{\prime \prime }{}^{\left(0\right)}\\ \vdots & p^{\prime }{}^{\left(1\right)} & p^{\prime \prime }{}^{\left(1\right)}\\ p^{\left(W\right)} & \vdots & \vdots \\ 0_{1\times N} & p^{\prime }{}^{\left(W\right)} & p^{\prime \prime }{}^{\left(W\right)} \end{array}\right],$$
where † is the transpose operation,
(16)$$p^{\prime }{}^{\left(w\right)}=\left\{\begin{array}{cc} p{}^{\left(w\right)} & \mathrm{for}\mathrm{odd}\mathrm{values}\mathrm{of}w\\ 0_{1\times N} & \mathrm{for}\mathrm{even}\mathrm{values}\mathrm{of}w, \end{array}\right.$$
and
(17)$$p^{\prime \prime }{}^{\left(w\right)}=\left\{\begin{array}{cc} 0_{1\times N} & \mathrm{for}\mathrm{odd}\mathrm{values}\mathrm{of}w\\ p{}^{\left(w\right)} & \mathrm{for}\mathrm{even}\mathrm{values}\mathrm{of}w, \end{array}\right.$$

The received signal at *D* represented by (Equation 5) and (Equation 12) for the entire frame can written in a condensed form as [33]
(18)R=PG+Z,
where Z is the noise vector produced by alternately mixing the components of (Equation 7) and (Equation 14).

The ML solutions of the modulation Φ and channel coefficients G are acquired by optimizing the log-likelihood function [37,39]
(19)$$\left[\hat{\Phi },\hat{G}\right]=\arg\max_{\Phi ,G}\;\log\mathrm{Pr}\left(R|\Phi ,G\right),$$
where $\mathrm{Pr}\left(\circ |▹\right)$ is the probability density function of ∘ given ▹. Calculating the term of $\mathrm{Pr}\left(R|\Phi ,G\right)$ involves taking an average over all potential data symbols that correspond to the modulation format Φ as described by [34]
(20)$$\mathrm{Pr}\left(R|\Phi ,G\right)=\sum_{P(\Phi )}\mathrm{Pr}\left(R|P(\Phi ),G\right)\mathrm{Pr}\left(P\left(\Phi \right)\right),$$
and
(21)$$\mathrm{Pr}\left(R|P(\Phi ),G\right)=\frac{1}{{\left(\pi \sigma_{n}^{2}\right)}^{N}}\exp\left(\;-\;{∥R-P(\Phi )G∥}^{2}\;/\;\sigma_{n}^{2}\right),$$
where $\sigma_{n}^{2}$ is the noise variance. It is essential to point out that we attach Φ to the matrix P described in (Equation 15) to underline that the transmission matrix is reliant on the particular modulation scheme that is chosen. One notes that the precise ML solution is extremely difficult, if not virtually impossible, to implement in practice. This is due to the fact that it necessitates the computation of (Equation 20) over a massive number of data symbols for each conceivable value of Φ. The EM approach offers a repetitive and conceptually straightforward mechanism for optimizing the log-likelihood, which may be used to a variety of problems. There are two processes of the operation at the iteration ξ+1.

E-process of the EM algorithm quantifies the expected value of $\log\mathrm{Pr}\left(R|P(\Phi ),G\right)$ given the existing estimations of the parameters Φ and G. In particular, we create the following function [42]
(22)$$\begin{array}{c} F\left(\Phi ,G\left|\hat{\Phi }\left(\xi \right),\hat{G}\left(\xi \right)\right.\right)=E_{P(\Phi )}\left[\log\mathrm{Pr}\left(R|P\left(\Phi \right),G\right)\left|\hat{\Phi }\left(\xi \right),\hat{G}\left(\xi \right)\right.\right], \end{array}$$Here $E_{P(\Phi )}\left[\cdot \right]$ signifies the average of the information symbols given the modulation type Φ.The M-process comprises optimizing the expectations that have been established in (Equation 22) as [43]
(23)$$\left[\hat{\Phi }\left(\xi +1\right),\hat{G}\left(\xi +1\right)\right]=\arg\max_{\Phi ,G}F\left(\Phi ,G\left|\hat{\Phi }\left(\xi \right),\hat{G}\left(\xi \right)\right.\right).$$

The EM method cycles through repeated steps until it reaches a point of convergence [48]. We construct the following equation by combining (Equation 15) and (Equation 20) into (Equation 19) and removing the extraneous components, [49]
(24)$$F\left(\Phi ,G\left|\hat{\Phi }\left(\xi \right),\hat{G}\left(\xi \right)\right.\right)=2ℜ\left(R^{†}\Omega \left(\Phi \right)G\right)-{\left|G\right|}^{2}\Psi \left(\Phi \right),$$
where $ℜ\left(\cdot \right)$ and † are the real part and the conjugate transpose of a complex-valued vector, respectively, [48]
(25)$$\Omega \left(\Phi \right)=E_{P(\Phi )}\left[P\left(\Phi \right)\left|R,\hat{G}\left(\xi \right)\right.\right],$$
and
(26)$$\Psi \left(\Phi \right)=E_{P(\Phi )}\left[P^{†}\left(\Phi \right)P\left(\Phi \right)\left|R,\hat{G}\left(\xi \right)\right.\right].$$
We carry out the following process in order to detach the joint maximization challenge of Φ and **G** of (Equation 23) as follows [15]
(27)$${\hat{G}}_{\Phi }\left(\xi +1\right)=\Psi {\left(\Phi \right)}^{-1}\Omega {\left(\Phi \right)}^{†}R.$$
Here, we attach Φ into G to highlight the fact that G is updated for each conceivable value of Φ. Plugging ${\hat{G}}_{\Phi }\left(\xi +1\right)$ into (Equation 23), the ξ+1 update of Φ is found by optimizing the cost function shown below [34],
(28)$$\begin{array}{c} \hat{\Phi }\left(\xi +1\right)=\arg\max_{\Phi }\left\{2ℜ\left(R^{†}\Omega \left(\Phi \right){\hat{G}}_{\Phi }\left(\xi +1\right)\right)-{∥{\hat{G}}_{\Phi }\left(\xi \right)∥}^{2}\Psi \left(\Phi \right)\right\}, \end{array}$$
where $∥.∥$ represents the norm of a vector. The conclusive measurement of G is stated as
(29)$$\hat{G}\left(\xi +1\right)={\hat{G}}_{\hat{\Phi }\left(\xi +1\right)}\left(\xi +1\right).$$

## 4. Practical Discussion

It is essential to note the following practical implications:The issue then arises of how the matrices of Ω(Φ) and Ψ(Φ) are really computed in reality. Since the expectation function is linear, we produce the matrix Ω(Φ) by simply replacing each ingredient of P(Φ) with the relevant a posteriori expectation. Typically, the a posteriori expectation of the *m*th element of the *w*th vector $p^{\left(w\right)}$, ${\tilde{p}}^{\left(w\right)}\left(m,\Phi \right)$, given the modulation scheme Φ is produced as [48]
(30)$$\begin{array}{c} {\tilde{p}}^{\left(w\right)}\left(m,\Phi \right)=E\left[p^{\left(w\right)}\left(m,\Phi \right)\left|R,\hat{\Phi }\left(\xi \right),\hat{G}\left(\xi \right)\right.\right],\\ {\tilde{p}}^{\left(w\right)}\left(m,\Phi \right)=\sum_{\gamma \in \Phi }\gamma \mathrm{Pr}\left(p^{\left(w\right)}\left(m,\Phi \right)=\gamma \left|R,\hat{\Phi }\left(\xi \right),\hat{G}\left(\xi \right)\right.\right). \end{array}$$The *w*th vector ${\tilde{p}}^{\left(w\right)}$ is created by concatenating all of these samples. This process is repeated for all unknown information symbols with w=0, 1, ⋯,W−1. Therefore, the matrix Ω(Φ) is formed following the same way as reported in (Equation 15) with the use of ${\tilde{p}}^{\left(w\right)}$ instead of $p^{\left(w\right)}$.We are going to proceed with the widely held premise that the data symbols are not linked with one another. Therefore, the approximate computation for Ψ(Φ) is as follows [34]
(31)$$\begin{array}{ccc} \Psi (\Phi ) & \approx & E_{P(\Phi )}\left[P^{†}\left(\Phi \right)\left|R,\hat{G}\left(\xi \right)\right.\right]\times E_{P(\Phi )}\left[P\left(\Phi \right)\left|R,\hat{G}\left(\xi \right)\right.\right],\\ & \approx & \Omega^{†}\left(\Phi \right)\Omega \left(\Phi \right). \end{array}$$In light of (Equation 30), computing the probability of $\mathrm{Pr}\left(p^{\left(w\right)}\left(m,\Phi \right)=\gamma \left|R,\hat{\Phi }\left(\xi \right),\hat{G}\left(\xi \right)\right.\right)$ for each modulation scheme Φ is essential for the proposed recognition process. When we examine (Equation 4) and (Equation 11) more closely, we come to the conclusion that each information symbol is collected at the endpoint twice: once through the source-destination connection and once via the relay-destination route. As a result, the corresponding transmission structure may be thought of as being comparable to a convolutional encoder that only has a single memory component. A bank of *N* Bahl, Cocke, Jelinek, and RavivBCJR algorithms is used to create the optimum data detection, which is accomplished by employing a method that is similar to the one described in the earlier works of [38,39].In order to kick off the recognition and estimation processes, a small number of pilot symbols are included into the data symbols. As a result, the proposed approach is able to get preliminary estimations of the channel coefficient and the chosen modulation scheme. It is generally known that the accuracy of the evaluation will improve in proportion to the number of utilized pilots. Nevertheless, extending the length of the pilot symbols not only reduces the amount of energy that is allocated for the transfer of information but also slows down the data rate. As a consequence of this, the ratio between the specified amount of pilot symbols and the amount of the information symbols has to be sustained at a level that is as low as possible. This is accomplished by using the method that has been described, which refines the original estimations via a process of iterative improvement. We make use of the output of the demodulation procedure to compute the a posteriori expectations of the information symbols, which are then applied to the proposed recognizer and estimator as if they were pilot symbols.It is worth noting that the quality of modulation recognition is strengthened by using a powerful error-correcting code to provide accurate feedback to the proposed technique. Furthermore, increasing the number of pilot symbols speeds up the suggested method’s convergence.The computational cost of the proposed AMR approach is examined by computing the number of needed floating point operations (fp) [15,50]. Multiplying two complex numbers needs 6 fp, whereas adding them only involves two fp. The study in the previous section shows that the required number of fp per iteration is roughly stated as 20WN. Assuming W=10 and N=100 with a processor speed of 10 Terafp per second, the processing time is 2 nanoseconds, which is sufficient for practical uses.

## 5. Simulation Results

Simulations have been implemented to assess how effectively the offered strategy behaves. The following is a list of the TCRS system’s parameters, unless anything else has been provided. The transmission frame is made up of W=25 packets. Each packet contains N=100 data symbols. The pilot symbols take up 7 percent of each frame. Quadrature amplitude modulation (QAM) and phase shift key (PSK) modulation schemes with a modulation order M= 2 to 256 were taken into account.A convolutional code was used, with a constraint length of 5 and a rate of 0.5. Each channel path was described as a zero-mean complex-valued Gaussian random variable, with variance being computed as [51,52,53]
(32)$$E\left[{\left|g_{\iota \kappa }\right|}^{2}\right]={\left(\frac{u_{SD}}{u_{\iota \kappa }}\right)}^{\varsigma },$$
where $\iota \in \left\{S,R_{1},R_{2}\right\}$, ς denoted to the path-loss exponent, $u_{\iota \kappa }$ was the separation between nodes ι and κ, and $u_{SD}$ was the separation between nodes *S* and *D*. For simulation reasons, we assign ς=3.4, $u_{SD}=1$, $u_{R_{1}D}=0.55$, and $u_{R_{2}D}=0.45$. Here, all distances were normalized to $u_{SD}$. As mentioned in the preceding mathematical advancements, the suggested technique is general in the sense that it can be applied to any other channel type. An evaluation metric for the proposed recognizer was the average probability of correct recognition, $A_{r}=\frac{1}{V}\sum_{\Theta }\mathrm{Pr}\left(\Phi =\Theta |\Theta \right)$, where Θ is a member of the set of modulation types and *V* is the cardinality of the modulation set under consideration. For example, we assume that the transmitter uses one of following modulation schemes, QPSK, 8-PSK, 16-QAM, and 64-QAM. Therefore, $A_{r}$ is described as $A_{r}=1/4\left(\mathrm{Pr}\left(\Phi =\mathrm{QPSK}|\mathrm{QPSK}\right)+\mathrm{Pr}\left(\Phi =8-\mathrm{PSK}|8-\mathrm{PSK}\right)+\mathrm{Pr}\left(\Phi =16-\mathrm{QAM}|16-\mathrm{QAM}\right)+\mathrm{Pr}\left(\Phi =64-\mathrm{QAM}|64-\mathrm{QAM}\right)\right)$. Here Pr(Φ=QPSK|QPSK) is defined as the probability of recognizing QPSK modulation scheme at the receiver if the allocated modulation scheme at the transmitter is QPSK. The same definition is applied to other modulation schemes.

Figure 2 illustrates $A_{r}$ performance of the offered recognizer as a function of symbol-to-noise ratio, SNR, at iterations 1–6. Even at high SNR values, the performance during the first iteration of the recognition process is insufficient since it is initiated by a limited number of pilots. As the number of iterations grows, there is a steady rise in the level of performance achieved. This is because the suggested recognizer is able to make use of the soft information that is supplied by the data detector, which becomes more robust with each repetition. After the sixth round, there is no distinguishable boost in the system’s performance. Because the vast majority of actual applications need for recognition accuracy of at least 90 percent, the minimum operational SNR would need to be more than 9 dB. In order to facilitate comparisons, we have also shown the recognition performance attained by the use of the methods outlined in [40,41]. The outcomes demonstrate that the proposed technique outperforms those methods, which are unable to provide adequate recognition performance despite using high SNR settings. The reason is that the methods of [40,41] follow a relaying strategy where the receiver sends back the decoded frame if it’s received successfully. When the relays are unable to understand the received signal because of the channel conditions, the modulation recognition performance worsens.

Figure 3 provides $A_{r}$ performance of the suggested recognizer in a variety of scenarios at iteration 6. In the first scenario, we assume that the channel estimate is accurate and we treat all of the data symbols as if they were training symbols. This acts as an upper-bound for the recognizer that is being suggested. In the second scenario, we use a few number of pilots to kick off the proposed modulation recognizer on the assumption that the channel estimate is accurate. The third scenario takes into account the possibility of performing channel estimate as mentioned in (Equation 29) and also executing modulation recognition as indicated in (Equation 28). The findings reveals that the offered EM-based recognizer achieves a level of recognition performance comparable to that of the first and second scenarios. This provides more evidence that the iterative structure that was described is successful.

Figure 4 depicts the impact of the number of pilots relative to the frame size at different values of SNR during iteration 6. It is well realized that the effective initialization is vital for EM-based schemes to run properly. The results indicate that the iterative recommended recognizer requires a ratio of at least 0.07 of the number of pilot symbols to frame size to achieve convergence. It is worth noting that the least number of pilots necessary to generate appropriate first estimations may vary depending on the system settings. In reality, the designer may put our estimate technique through its paces by varying system and connectivity settings, finally establishing the bare minimum number of pilots required to meet the criterion for every given parameter choice. After that, the findings are recorded into look-up databases so that they can be deployed on the hardware. The designer also has the option of arbitrarily selecting a sufficient number of pilots that are suited for a broad variety of system and connectivity characteristics. Table 2 shows the relationship between pilot to frame size ratio, recognition accuracy, and number of iterations necessary to converge at SNR = 14 dB. The results show that increasing the number of pilots improves the rate of recognition and reduces the number of iterations needed to reach convergence. Even with a considerable number of iterations, a small number of pilots does not result in adequate recognition performance. Furthermore, a massive proportion of pilots results in flawless recognition performance with a minimal amount of iterations.

Throughout the preceding discussions, we used the assumption that the receiver estimates and corrects the clock-timing and frequency mismatches in a precise manner. The next investigations evaluate how sensitive the recommended recognition approach is to these parameters. Figure 5 and Figure 6 demonstrate, respectively, how the time offset τ and the frequency offset ν impact the recognition performance at different values of SNR during iteration 6. Here τ is normalized to the sampling duration and is represented as a two-way channel [1−τ,τ] [46] and ν is normalized to the carrier frequency. These outcomes show that the suggested approach is relatively resistant to τ and delivers adequate performance over a broad range of ν.

## 6. Conclusions

We explored a decision feedback receiver architecture of a destination node for amplify and forward two-path consecutive relaying systems throughout this study. We developed a combined approach for iterative modulation recognition and channel estimation for channel impulse responses across source, relay, and destination nodes. It was shown that the maximum likelihood solution of the problem under consideration can be implemented via employing an expectation-maximization procedure. The acquisition of the preliminary estimations was generated by using a few pilot symbols. The suggested recognizer and estimator iteratively communicated and traded soft information with the data detector to achieve better overall performance. According to the simulation results, an outstanding recognition performance was shown to be exhibited in a variety of functional conditions. The recommended approach converged within six iterations and achieved flawless recognition performance at a signal-to-noise ratio of 14 dB. In addition, the minimum pilot-to-frame-size ratio required to conduct the iterative technique properly is 0.07. Finally, the proposed method was largely impervious to time offset and offered good performance across a wide range of frequency offset.

## Figures and Tables

**Figure 1 sensors-22-06022-f001:**
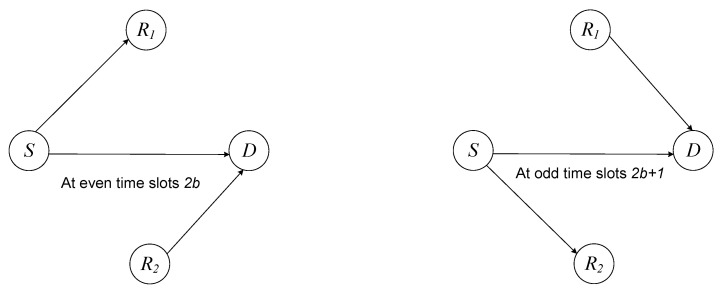
The TPSR system under consideration.

**Figure 2 sensors-22-06022-f002:**
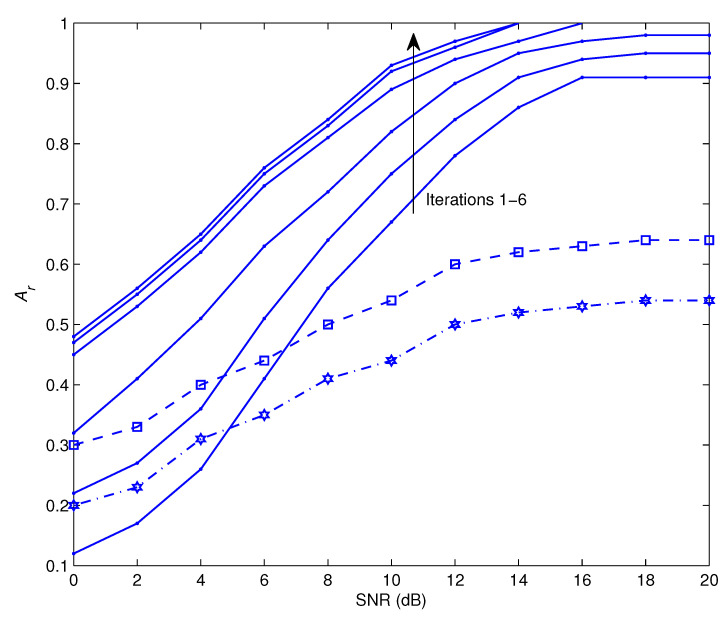
$A_{r}$ as a functions of SNR at different iterations. The proposed algorithm is illustrated by the solid lines, the algorithm of [40] is displayed by the dashed lines, and the algorithm of [41] is indicated by the dot-dashed lines.

**Figure 3 sensors-22-06022-f003:**
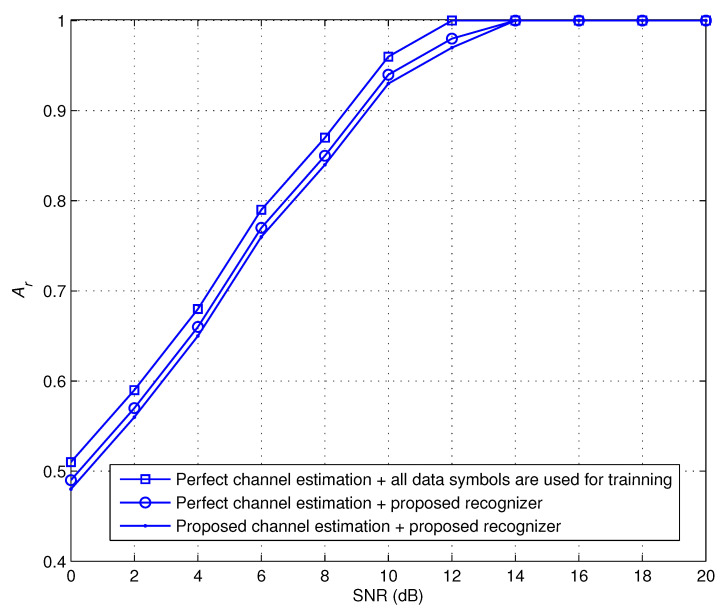
$A_{r}$ performance at different scenarios.

**Figure 4 sensors-22-06022-f004:**
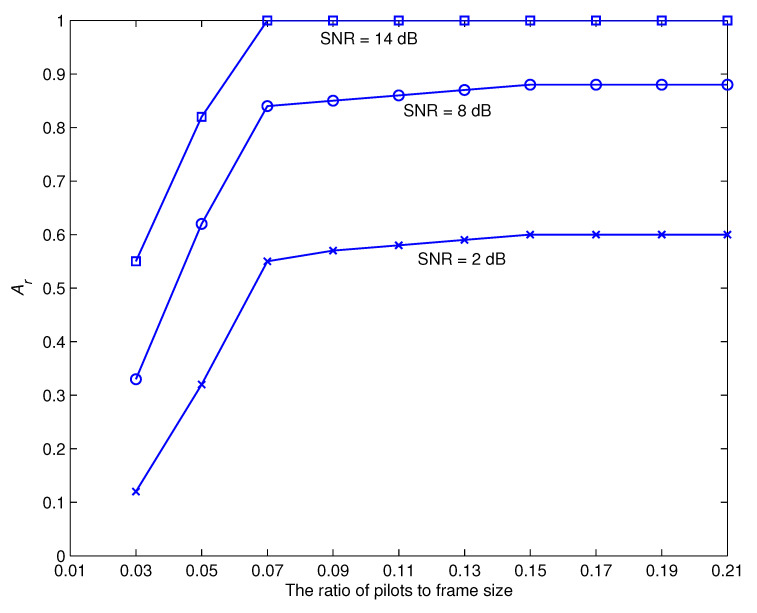
The influence of the number of pilots relative to the frame size with six iterations.

**Figure 5 sensors-22-06022-f005:**
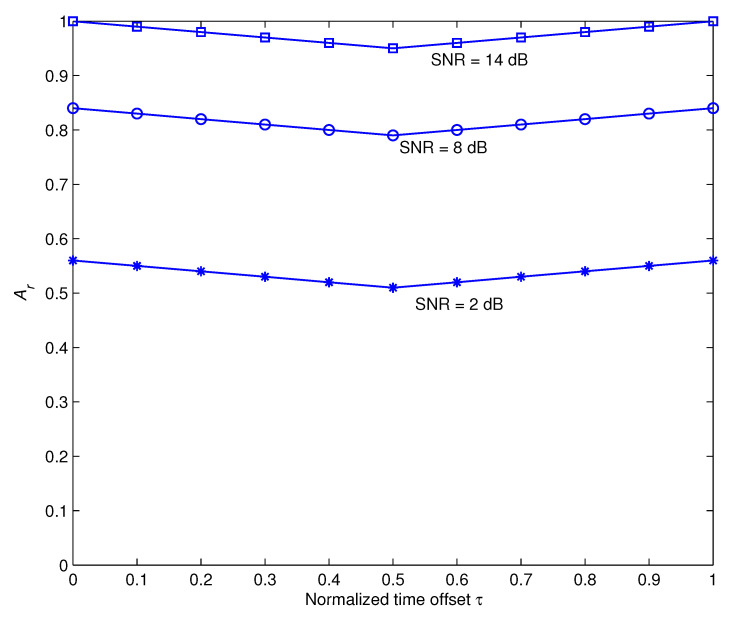
The impact of normalized time offset τ on the recognition performance.

**Figure 6 sensors-22-06022-f006:**
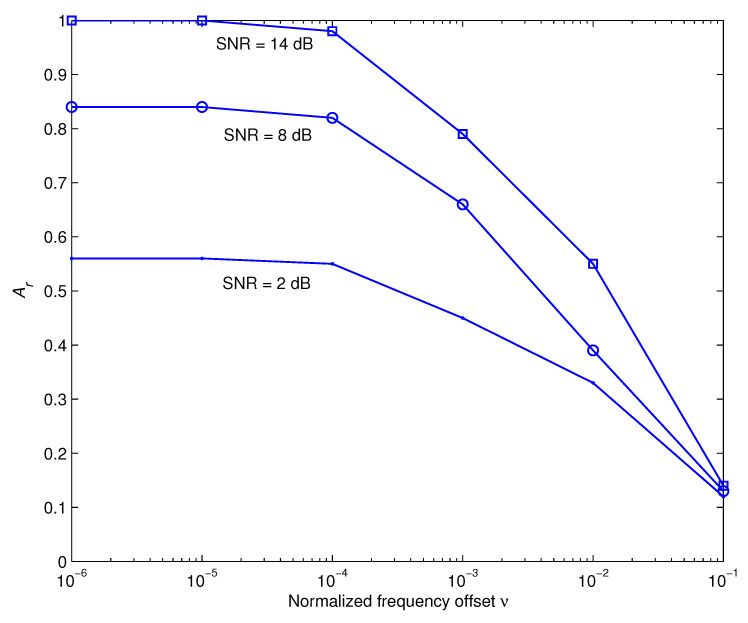
The impact of normalized frequency offset ν on the recognition performance.

**Table 1 sensors-22-06022-t001:** List of abbreviations.

Abbreviation	Definition
AMR	Automatic modulation recognition
AAF	Amplify-and-forward
TCRS	Two-path consecutive relaying systems
MIMO	Multiple-input multiple-output
ML	Maximum-likelihood
EM	Expectation-maximization
DA	Data-aided

**Table 2 sensors-22-06022-t002:** The relationship between pilot to frame size ratio, recognition accuracy, and number of iterations necessary to converge at SNR = 14 dB.

Pilot to Frame Size Ratio	Recognition Accuracy	Iterations Necessary to Converge
0.03	0.72	14
0.05	0.91	10
0.07	1	6
0.09	1	4
0.11	1	3
0.13	1	2

## Data Availability

Not applicable.

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
