# Peer review of "Decision Feedback Modulation Recognition with Channel Estimation for Amplify and Forward Two-Path Consecutive Relaying Systems"

_sensors, 2022, doi:10.3390/s22166022_

Round 1

Reviewer 1 Report

Please have a look on the attached reviewer report

Paper needs major revisions

Reviewer 2 Report

The manuscript “Decision Feedback Modulation Recognition With Channel Estimation for Amplify And Forward Two-Path Successive Relaying Systems” presents and examines the AMR problem in the context of amplify-and-forward two-path consecutive relaying systems Authors leverage the property of data redundancy associated with amplify-and-forward two-path consecutive relaying systems signals to design an iterative modulation recognizer via an expectation-maximization procedure. The proposed method incorporates the soft information produced by the data detection process as a priori knowledge to generate the a posteriori expectations of the information symbols – employing training symbols. The presented idea is interesting and developmental. 

1.        Title

The title should reflect its main idea, e.g., a specific approach, method, scenario, novelty aspect, etc. Generally, the title of the reviewed paper reflects well the paper's contribution.

2.        Abbreviation

The authors explain all of the used abbreviations. There is no one missing:

3.        Content

The introduction provides sufficient background and includes relevant references. The research design is appropriate, and the method is adequately described. The conclusions are supported by the results, which are clearly presented.

English language and style are acceptable.

Simulation results

The considered noise used in the research on the effectiveness of the developed method was Gaussian noise. It is the simplest solution in mathematical models. Radio channel path defined as zero-mean complex-valued Gaussian random variable is suitable as a 1st algorithm evaluation. What about the more realistic situation for Rayleigh or Rician fading channels? I’ve missed the more real-life simulation scenarios.

4.        References

The literature sources are correctly selected and up-to-date concerning the topic under consideration

Reviewer 3 Report

The authors claim that “The proposed recognizer incorporates the soft information produced by the data detection process as a priori knowledge to generate the a posteriori expectations of the information symbols, which are employed as training symbols. The proposed algorithm additionally involves the development of an estimate of the channel coefficients as a secondary activity.” The simulations should corresponded demonstrate how the proposed recognizer and estimator help the relay scheduling. The authors also need clearly present the physical meaning of Ar in the simulation section.

Round 2

Reviewer 1 Report

Reviewers replied effectively to all comments. The paper is suitable for publication.

Author Response

Thank you for your positive opinion about the work.